# Design and Performance of Polyurethane Elastomers Composed with Different Soft Segments

**DOI:** 10.3390/ma13214991

**Published:** 2020-11-05

**Authors:** Xin Jin, Naisheng Guo, Zhanping You, Yiqiu Tan

**Affiliations:** 1Transportation Engineering College, Dalian Maritime University, Dalian 116026, China; jinxinzzz@126.com; 2Department of Civil and Environmental Engineering, Michigan Technological University, Houghton, MI 49931-1295, USA; zyou@mtu.edu; 3School of Transportation Science and Engineering, Harbin Institute of Technology, Harbin 150090, China; yiqiutan@163.com

**Keywords:** thermoplastic polyurethane elastomers, different soft segments, microstructure characterization, mechanical properties

## Abstract

Thermoplastic polyurethane elastomers (TPUs) are widely used in a variety of applications as a result of flexible and superior performance. However, few scholars pay close attention on the design and synthesis of TPUs through the self-determined laboratory process, especially on definite of chemical structures and upon the influence on properties. To investigate the properties of synthesized modifier based on chemical structure, firstly each kind of unknown structure and composition ratio of TPUs was determined by using a new method. Furthermore, the thermal characteristics and mechanical properties of modifiers were exposed by thermal characteristics and mechanics performance tests. The experimental results indicate that TPUs for use as an asphalt modifier can successfully be synthesized with the aid of semi-prepolymer method. The linear backbone structure of TPUs with different hard segment contents were determined by micro test methods. The polyester-based TPUs had thermal behavior better than the polyether-based TPUs; conversely, the low temperature performance of polyether-based TPUs was superior. Most importantly, it was found that the relative molecular mass of TPUs exhibited a weak effect on the mechanical properties, whereas the crystallinity of hard segment showed a significant influence on the properties of TPUs.

## 1. Introduction

Thermoplastic polyurethane (TPU) is an emerging organic polymer material. Structurally, TPU is also a kind of block polymer, which is generally composed of a flexible long chain of oligo-polyols to form a soft segment and a hard segment with diisocyanate and chain extender. The hard and soft segments may alternate arrangement to form the structure of repeating units [1,2]. In general, the main kinds of TPUs can be divided into polyester, polyether, polyolefin, etc. Otherwise, TPUs can be subdivided into 2,4-tolylene diisocyanate (TDI)-based TPUs and type 4,4′-diphenylmethane diisocyanate (MDI)-based TPUs according to the different diisocyanates [3]. TDI and MDI are the main raw materials for synthesizing TPUs, which are quite different from each other in the subdivision utility due to the differences between the structure and performance [4]. The molecular structure of MDI exhibits a better symmetry type compared with TDI, due the NCO in the opposite position of the two benzene rings of MDI. Moreover, the methylene between the benzene rings can effectively reduce the abrasion resistance of the benzene rings; therefore, MDI has excellent flexibility and the molecular chain is relatively neat and easy to be crystallized, which makes the TPUs using MDI as the hard segment show higher modulus and lower mechanical properties. By contrast, fewer and fewer researchers are using TDI to synthesize TPUs as a result of its extreme toxicity, which is far less extensive than MDI [5]. In 1959, MDI was introduced to China [6], and its consumption exceeded TDI in 1984 [7]. In recent years, as the global demand for TPUs is rising, the output of MDI has increased rapidly. The products produced based on MDI have penetrated into almost all products in the TPU field, owing to the TPUs synthesized by MDI having excellent comprehensive properties to meet environmental protection requirements.

Currently, the TPUs synthesized using MDI are divided into polyester-based and polyether-based TPUs. The similarity between them is that chemical reactions exist between the molecules, namely the hydrogen bonding or light cross-linking between macromolecular chains. These cross-linked structures undergo a reversible chemical change during heating or cooling. In other words, the intermolecular forces will gradually weaken in the molten state, and the strong intermolecular forces will be restored after cooling, which can make TPUs recover the properties of the original solid. TPUs have attracted much attention due to their unique properties, such as abrasion resistance, ozone resistance, low temperature resistance, chemical resistance, etc. In view of these characteristics, TPUs have been widely used in the fields of national defense technology, electronic technology, engineering construction, medical equipment, etc. [8,9,10,11,12]. The hard segment includes MDI and small molecule diol (chain extender), while polyol is used to form the soft segment [13]. To fully develop and utilize TPUs, the monomer structure, the hard segment content, and the molecular weight can be changed to meet the specification requirements in practical use. On the other hand, the performance of TPUs can be modified within a wide range by adjusting the types and ratios of the soft segment, hard segment, and chain extender [14,15]. Thus, it is very important to determine the optimal hard segment content and the appropriate soft segment structure in the synthesis process of TPUs, which have great significance for guiding the development of two-component TPU resins with low foaming, high strength, and high modulus, which can be used as road engineering materials.

Over the decades, many researchers have investigated the synthesis method of TPUs considering the mechanical function and heat performance. Król [16] focused on TPUs synthesized by polyester and polyether with the different hard segment contents, and the results show that the mechanical properties of polyester TPUs are considered better than those of polyether TPUs using the same preparation method. Li et al. [17] found that the interaction between N–H and CO groups showed a remarkable variation with increasing temperature and revealed the endotherms origin from the intermolecular interactions level and the specific orders of temperature response during hydrogen bonds dissociation. They also presented a schematic evolution containing factors influencing the endotherms behaviors. Rosu et al. [18] revealed that the TPUs synthesized from MDI, PEA, and BD contained at least four stages of thermal decomposition in inert atmosphere. The effect of temperature on the thermal decomposition of PU initiated with macromolecules scission as well as dehydration reactions of glycols, and a simultaneous and partial destruction of ester linkages from hard and soft segments in the TPU structure appeared. Barrioni et al. [19] used hexamethylene diisocyanate and glycerol as the hard segment and poly(caprolactone) triol and low-molecular-weight poly(ethyleneglycol) as the soft segment without the use of a catalyst. A highly connected network with a homogeneous PU structure was obtained from crosslinked bonds. The films showed amorphous structures, high water uptake, hydrogel behavior, and susceptibility to hydrolytic degradation. However, these studies were inconsistent with the actual structures because they failed to consider the relationship between material composition and physical and mechanical properties. There have been few studies on how to determine the sequence structure of TPUs based on different soft segments. Generally, the isocyanates react with polyols to form TPUs. Nevertheless, different types of polyols can be used to synthesize TPU resin with different soft structures, and the properties of the TPUs obtained are also different. Therefore, by comparing the properties of TPUs with different soft segment structures, and the influence of soft segment structure on the properties of MDI-based TPU can be explored, which is of great significance for improving autonomation and developing new products with high-performance, as well as exploring the application of TPUs in the engineering domain.

Herein, polyester-based and polyether-based TPUs were successfully designed and synthesized as raw materials, which were obtained by polytetramethylene ether glycol (PTMEG), poly(1,4-butylene adipate) (PBA), methylene diphenyl diisocyanate (MDI), and 1,4-butylene glycol (BDO). The TPU samples with different soft segments were prepared by the semi-polymerization method, in which the polyester polyols and polyether polyols with the equal molecular weight were used as soft segments, while MDI and BDO were used as hard segments. It is also worth mentioning that the effects of the composition, amounts, and molecular weights of the polyether polyol on the performance of the TPUs were investigated. This paper mainly describes the methods to deduce the microstructure of the polyester-based and polyether-based TPUs in which the thermal properties and mechanical properties were considered. Moreover, the synthesis mechanisms of TPUs were revealed in order to explore the new deduction technology of chemical structure and provide a theoretical basis for the development of TPUs for engineering.

## 2. Materials and Methods

### 2.1. Materials

The poly(1,4-butylene adipate) (PBA, Mn = 2000) used in the study was acquired from Yantai Huaxin Polyurethane Co., Ltd., Yantai, China. Polytetramethylene ether glycol (PTMEG, Mn = 2000), methylene diphenyl diisocyanate (MDI, Mn = 250.25) and 1,4-butylene glycol (BDO, Mn = 90.12) were supplied by Shanghai Aladdin Bio-Chem Technology Co., Ltd., Shanghai, China. Tetrahydrofuran (THF) was purchased from Sinopharm Chemical Reagent Co., Ltd., Shenyang, China. Dimethyl-d6 sulfoxide (d6(DMSO)) was supplied by Meryer Chemical Technology Co., Ltd., Shanghai, China.

### 2.2. The Equation Design of TPUs

TPUs can be plasticized by heating. TPUs were prepared using n, N-dimethylformamide (DMF) which was supplied by Beijing Borunlaite Science& Technoogy Co., Ltd., Beijing, China as solvent. In the TPU molecular structure, the soft segment with a large proportion accounts for 50–90% and the hard segment accounts for 10–50%. There are many kinds of raw materials for synthesizing TPUs, and the chemical reactions involved are complicated. The quality guarantee of TPUs comes from strict raw material ratio control (see Figure 1).

Usually, the hard segment of TPUs is a diisocyanate with cyclic, tight, symmetrical cores to make its particle dense, which can enhance the physical properties. Therefore, methylene diphenyl diisocyanate (MDI) was chosen as a hard segment in this study. The molecular weight of the oligomeric diol used in the soft segment is generally 500–4000 (1000–2000 is usually selected), hence polytetramethylene ether glycol (PTMEG) or poly(1,4-butylene adipate) (PBA) was used as the soft segment of TPUs. Chain extenders with small molecular weights and no substituents (generally lower glycols are preferred) facilitate the aggregation of hard segments and urethane groups. Consequently, 1,4-butanediol was selected as the chain extender (BDO).

The hard segment content (*C*_h_) is the mass fraction of hard segment in TPUs, and it is the mass sum of diisocyanate and small molecule diol. The isocyanate index (*r*) refers to the equivalent number ratio of diisocyanate to sum of the oligomeric diol and small molecule diol, or the ratio of NCO/OH. The four raw materials used in this study were difunctional groups; thus, NCO/OH is the molar ratio given by
(1)$$C_{h}=\frac{W_{i}+W_{d}}{W_{i}+W_{d}+W_{g}}$$
(2)$$r=\frac{n_{\mathrm{NCO}}}{n_{\mathrm{OH}}}\;=\;\frac{\frac{W_{i}}{M_{i}}}{\frac{W_{d}}{M_{d}}+\frac{W_{g}}{M_{g}}}$$

According to Equations (1) and (2), Equation (3) can be obtained:(3)$$C_{h}=\frac{r\times \frac{W_{i}}{M_{i}}\times M_{i}+\left(1+\frac{r\;\times \;M_{i}}{M_{d}}\right)\times W_{d}}{\left(1+\frac{r\;\times \;M_{i}}{M_{d}}\right)\times W_{d}+\left(1+\frac{r\;\times \;M_{i}}{M_{g}}\right)\times W_{g}}$$
where *W*_i_ is the mass of MDI; *W*_d_ is the mass of BDO; *W*_g_ is the mass of PBA or PTMEG; *M*_i_ is the relative molecular mass of MDI; *M*_d_ is the relative molecular mass of BDO; *M*_g_ is the relative molecular mass of PBA or PTMEG; *C*_h_ is the hard segment content in TPU; and *r* is the isocyanate root index.

PBA and PTMEG with a molecular weight of 2000 were selected as the soft segment of the TPUs. The hard segment content was designed to be 20%, 30% or 40%, and the *r* was controlled as 0.95, 1 or 1.05, respectively. The amount of PBA/PTMEG was 100 g, the mass of BDO and MDI was calculated according to Equations (2) and (3), respectively. The 18 kinds of synthetic TPUs are shown in Table 1 and Table 2, respectively.

### 2.3. Sample Preparation

In this study, the semi-prepolymer method was used to synthesize TPUs, and the preparation process is shown in Figure 2.

### 2.4. Measurements

#### 2.4.1. Fourier Transform Infrared (FTIR)

The functional groups of reaction mixture characteristic groups were identified by FTIR characteristic adsorption peaks, which were tested by IRT-100 infrared spectrometer (SHIMADZU, Japan), Attenuated Total Reflectance (ATR) mode. The spectra were collected through 32 scans with a spectral resolution of 4 cm^−1^ for a spectral range from 400 to 4000 cm^−1^.

#### 2.4.2. Elemental Analysis (EA)

The elemental analyses were performed using a Vario EL III analyzer (Elementar, Germany) to obtain element content of TPUs. The hard segment of the TPUs synthesized in this study was composed of MDI and BDO, which were copolymerized with the soft segment (PBA or PTMEG) to form the TPU block copolymers. The hard segment content was set to A, and the soft segment content was set to B; then, A + B = 100. The elemental analysis was used to calculate A (Equation (4)) [20,21,22]:(4)$$A=\frac{E_{\mathrm{AB}}-E_{B}}{E_{A}-E_{B}}\times 100\%$$
where *E*_AB_ is the percentage content of C, H or N element in AB was measured by elemental analyzer. *E*_A_ and *E*_B_ are the theoretical values of *A* and *B*, respectively. Because *A* + *B* = AB, when the percentage of an element in the composition was zero (*E*_B_ = 0), the above equation can be simplified as follows:(5)$$A=\frac{E_{\mathrm{AB}}}{E_{A}}\times 100\%$$

#### 2.4.3. Gel Permeation Chromatography (GPC)

Molecular weight and distribution of samples were performed at 30 °C by using tetrahydrofuran (THF) as a mobile phase on a PL-GPC50 (Polymer Laboratories, UK) apparatus equipped with a Malvern refractive index detector at an elution rate of 1 mL/min, while monodispersed polystyrene (PS) solution was used as the standard, and each sample was prepared at a concentration of 1.5% (g/mL).

#### 2.4.4. Nuclear Magnetic Resonance (NMR)

^1^H and ^12^C NMR spectra were recorded on an AVANCE III HD 500 (BRUKER OPTICS, Germany) NMR spectrometer operated in the Fourier transform mode with an operating frequency of 400 MHz at 25 °C. Dimethyl-d_6_ sulfoxide (d_6_(DMSO)) was used as the solvent. The solution was measured with tetramethyl silane as an internal reference.

#### 2.4.5. Vicat Softening Temperature (VST)

The physical and mechanical properties of TPUs subjected to the thermal conditioning was evaluated by WXB-300C (GOTECH, China) tester using silicone oil as warming medium. Each sample (80 mm × 10 mm × 4 mm) was measured under a loading weight of 1000 g heating from 25 °C to Vicat softening temperature at a rate of 50 °C/h in terms of GB/T 1633–2000.

#### 2.4.6. Thermogravimetric (TG)

Thermal stability was performed using Q 50 (TA instruments, USA) thermogravimetric instrument. Each sample was loaded into a Pt pan, and a sapphire plate (10 mg) was placed in a separate Pt pan as the reference. The mass of the sample was between 3 and 5 mg; the heating rate was 20 °C/min, from 25 to 800 °C; and there was a nitrogen (N_2_) atmosphere.

#### 2.4.7. Differential Scanning Calorimetric (DSC)

DSC experiments were represented by Q 20 (TA instruments, USA) instrument. A scale (1/10,000 g) was adopted to measure the weight due to the high precision requirement of the sample that is not more than 5 ± 1 mg. Under purge gas in DSC, the sample was heated from room temperature to 200 °C at a heating rate of 10 °C/min; kept at a constant temperature for 2 min; cooled to −75 °C at a rate of 10 °C/min; maintained at a constant temperature for 1 min; and then heated to 225 °C at a heating rate of 10 °C/min. Finally, the DSC curves were obtained. The glass transition was analyzed by the DSC result; the glass transition temperature (*T*_g_) can be obtained by the DSC testing curves of TPUs.

#### 2.4.8. Physical and Mechanical Properties

Hardness tests were carried out with a HT-6510C (Shuangxu, China) manual Shore A durometer. The sample needs to meet the specification of 50 mm × 50 mm × 4 mm. Measurements were taken after 15 s at 25 °C in accordance with equivalent China Standard GBT1698–2003 to ASTM standard D 2240, taking 10 measurement points and calculating their average, which was regarded as the hardness value. Impact strength tests were performed on the XJV-22 (GOTECH, China) beam impact testing machine using the rectangular samples at room temperature in accordance with the equivalent ASTM D256-97 standard to GB/T 1043.1–2008 standard. Five samples of every group were measured, and the averages were obtained. Tensile strength, tensile stress at 300%, and elongation at break were tested by 5900 (Instron, Norfolk County, MA, USA) universal electronic tensile testing machine. The stretching rate was 50 mm/min in accordance with equivalent ASTM D638-14 standard to GB/T 1040.2-2006 standard at 25 °C. The measured length of dumbbell specimen (gauge length) was 50 mm, and the the average of five repeated tests was obtained. The tear strength test was also employed using universal electronic tensile testing machine. Each sample was a right-angle type (GB530-81) with the thickness of 2 mm. The testing result was achieved from five repeated tests and the test method was conducted on the basis of equivalent ASTM D624 standard to GBT 529-2008.

## 3. Results and Discussion

### 3.1. Chemical Mechanisms

#### 3.1.1. Fourier Transform Infrared (FTIR)

Spectral analysis of MDI, BDO, PBA, and PTMEG conducted by using FTIR are shown in Figure 3a. The MDI spectrum showed a strong asymmetric stretching vibration at 2265 cm^−1^ for –NCO. Moreover, a few weak symmetrical stretching vibrations ranging from 1375 to 1395 cm^−1^ were assigned to –NCO [23,24], whereas the spectrum exhibited a characteristic group of –CH at 2900 cm^−1^. Similarly, the intense broad bands at 1100 and 650–900 cm^−1^ were the vibrations of benzene ring and its isocyanate derivatives [25].

In addition, the FTIR spectrum of BDO is shown in Figure 3a. A remarkable broad band at 3291 cm^−1^ was credited to inter-molecular hydrogen bonding and the vibration stretching corresponding to –OH. The two bands of absorption at 2937 and 2867 cm^−1^ were attributed to vibration stretching of –CH. The spikes at 1734, 1457, and 1049 cm^−1^ represented carbonyl or ester groups, which are beneficial to the regular arrangement and close packing of the hard segments due to the symmetry of the chain extender molecules, making soft segments difficult to integrate the hard segment. Therefore, BDO, chosen as the chain extender, can improve micro-phase separation of the soft and hard segments of TPUs. To finalize the polymerization reaction, BDO was added in a second step [26].

The absorption peak in the spectrum of PBA is shown in Figure 3b. The peak around 3531 cm^−1^ corresponded to the vibration stretching of –OH. The symmetrical and asymmetrical stretching vibrations of –CH corresponded to CH_2_ at 2956 and 2875 cm^−1^ [27], respectively. In addition, the absorption peak at 1360 cm^−1^ corresponded to the symmetrical bending vibration of –CH in CH_3_. The band at 1250 cm^−1^ was the characteristic absorption of C–O–C stretching vibrations in esters and ethers. An absorption peak at 1725 cm^−1^ was attributed to the vibration stretching of C=O [28]. Two absorption peaks at 1464 and 1419 cm^−1^ were C=C vibrational stretching of benzene ring skeleton. The absorption peaks at 1150, 1060, and 965 cm^−1^ were attributed to the in-surface bending vibration of =CH on the benzene ring. Besides, the absorption peak at 731 cm^−1^ was also the bending vibration of =CH on the benzene ring [29]. The tested sample contained many hydroxyl and ester groups due to the appearance of strong hydroxyl and ester groups.

The molecular structure of polyether polyols was characterized by containing hydroxyl –OH and ether bonds C–O–C [30]. As shown in Figure 3b, the characteristic urethane group band was shown at 3444 cm^−1^ in the FTIR spectrum, which was related to the overlapping of –OH. The peaks of –CH in CH_3_ asymmetric stretching vibration and symmetric vibration or –CH symmetric stretching vibration in CH_2_ appeared at 2861 and 2890 cm^−1^. The absorption peak at 1727 cm^−1^ was the stretching vibration of C=O. Additionally, the absorption peak values in the range of 1300–1500 cm^−1^ were due to in-plane bending vibration and out-of-plane bending vibration of –OH, while the vibration caused by –CH deformation was also at this phase. Nevertheless, the other peaks of absorption in the FTIR spectrum of PTMEG were designated as stretching vibration of CH_2_ at 1368 cm^−1^ and irregular stretching vibration of C–O–C at 1250 cm^−1^ [31,32,33]. The absorption peaks at 1161 and 1068 cm^−1^ were caused by the stretching vibration of C–O connected to carbon atoms [34].

The FTIR spectra of the polyester-based TPUs is shown in Figure 4. It can be seen that all samples exhibited the identical peak type and peak position, whereas the size of the peaks indicated a slight variation. Two strong absorption peaks at 2956 and 2817 cm^−1^ were attributed to stretch vibrations of CH_2_ and CH_3_, respectively. The spectrum exhibited the characteristic bands of polyester-based TPUs at 1722 and 1538 cm^−1^ due to the C=O stretching vibration (amide I band), N–H (amide II band) bending vibration, and C–N (amide III band) stretching vibration, respectively. The skeletal vibration of CH_3_ in the aromatic ring at 1380 cm^−1^ can be coupled with the absorbance between 816 and 712 cm^−^^1^, which was the most significant characteristic of C–H out-of-plane bending vibration in 1,4-disubstituted aromatic ring [35].

As shown in Figure 5, the strong absorption peaks appeared at 2948 and 2853 cm^−1^, which were caused by telescopic vibrations of CH_2_ and CH_3_, respectively. The observed peaks within such spectral region were designated as a strong absorption peak occurring at 1728 cm^−1^, which was assigned to the stretching vibration of C=O in the ester carbonyl group or the stretching vibration of C=O (amide I band). When the hard segment content was more than 20%, a new peak occurred at 1700 cm^−1^, which was assigned to C=O of the isocyanurate trimer (another peak existing in the range of 1408–1430 cm^−1^ was also attributed to isocyanurate trimer). The N–H deformation vibration appeared from 1520 to 1560 cm^−1^ (amide II band). The other peaks of absorption in the FTIR spectrum of TPUs were designated as 1227–1233 cm^−1^ for deformation vibration of –OH, 1060–1150 cm^−1^ for C–O–C stretching of aliphatic bond, 1110–1080 cm^−1^ for C–O stretching vibration in the ether, and 816–712 cm^−1^ for –CH bending on the benzene ring.

In Figure 4 and Figure 5 regardless of polyester-based or polyether-based TPUs, the characteristic absorption peak of –NCO at about 2270 cm^−1^ disappeared after adding the BDO, which suggested that –NCO and –OH reacted completely. In addition, the characteristic urethane group band was shown at 3295 and 1536 cm^−1^ in the FTIR spectrum, which belongs to the stretching of N–H bond observed in the TPUs that showed the intermolecular hydrogen bonding in the –NH of TPUs, i.e. C=O at 1725 cm^−1^ (some scholars believe that it is incomplete hydrogen-bonded urethane carbonyl [36] or free urethane carbonyl [37]). The infrared spectrum showed that the preparation process used in this study can achieve the desired product.

#### 3.1.2. Elemental Analysis (EA)

The reaction type of TPUs is regarded as addition polymerization, which meets *A* + *B* = AB. The hard segment content can be calculated from the N element due to the lack of N element in the soft segment. Therefore, the hard segment was 8.24% after calculating the theoretical value of the N element, where *E*_A_ = 8.24%. The hard segment content in the TPUs can be obtained by Equation (5).

The EA results of the synthesized TPUs are listed in Table 3 and Table 4. It was found that the design values of the hard segment contents of the synthesized TPUs were similar to the measured values, indicating that the material design was reasonable and the synthetic method was feasible. Therefore, it was proved that the desired product was obtained.

#### 3.1.3. Gel Permeation Chromatography (GPC)

To exactly characterize the molecular weight (including number average molar mass ($\overset{¯¯}{M}$_n_), mass average molar mass ($\overset{¯¯}{M}$_w_), z-average molar mass ($\overset{¯¯}{M}$_z_), and viscosity average molar mass ($\overset{¯¯}{M}$_v_)) of polymer, as well as give the statistical average of molecular weight, the molecular weight distribution index (D) must be provided. The molecular weight and distribution of TPUs were characterized by GPC, as shown in Table 5 and Table 6.

The molecular weight is a key index affecting the properties of TPUs. Table 5 shows that, for identical hard segment content, the molecular weight and *D* increased gradually with increasing *r*. When the *r* was constant, the molecular weight decreased gradually with the increase of the hard segment content, while the variation of *D* was not obvious. When the hard segment content was 20% and *r* = 1.05, $\overset{¯¯}{M}$_n_, $\overset{¯¯}{M}$_w_, $\overset{¯¯}{M}$_z_, and $\overset{¯¯}{M}$_v_ of the polyester-based TPUs presented the largest values, namely 161,147, 486,596, 123,832, and 414,624, respectively, and the *D* showed the widest value (3.02). Since polymers are different from low molecular weight compounds, the wider is the molecular weight distribution, the greater is the polydispersity of the molecules. When the hard segment content was 40% and *r* = 0.95, $\overset{¯¯}{M}$*_n_*, $\overset{¯¯}{M}$_w_, $\overset{¯¯}{M}$_z_, $\overset{¯¯}{M}$_v_, and *D* of the polyether-based TPUs exhibited their minimum and the polydisperse degree of the molecules was the minimum with a narrow molecular distribution. It was found that the molecular weight of polyether-based TPUs was inversely correlated with the molecular weight distribution index.

As shown in Table 6, the polyester-based TPUs exhibited a similar fluctuating trend with the polyether-based TPUs. The molecular weight and molecular weight distribution index increased with the increase of *r* in terms of a certain hard segment content. When the hard segment content was 20% and *r* = 1.05, $\overset{¯¯}{M}$*_n_*, $\overset{¯¯}{M}$_w_, $\overset{¯¯}{M}$_z_, and $\overset{¯¯}{M}$_v_ of the polyester-based TPUs were also very similar to the polyester-based TPUs, namely 161,147, 486,596, 123,832, and 414,624, respectively. For the identical *r*, the size of molecular weights of TPUs with different hard segment contents were ranked as: *C*_h_ = 20% > *C*_h_ = 40% > *C*_h_ = 30%. Furthermore, when the hard segment content was 20% and *r* = 0.95, the molecular weight distribution was the minimum (1.89), showing that the polydisperse degree of molecules was the minimum. It was also found that the molecular weight of polyester-based TPUs gradually increased with increasing *r*, which meant that the molecular weight of TPUs increased with the increase of –NCO.

In conclusion, the chemical structure of both polyester-based and polyester-based TPUs with different hard segment contents can be deduced through FTIR, EA, and GPC. The synthetic chemical equation of the polyester-based TPUs using MDI and BDO as the hard segment and PBA as the soft segment is shown in Figure 6.

A hydroxyl-terminated PBA (C_12_H_22_O_6_) with a relative molecular mass of 2000 was used in this study. The relative molecular mass of the monomer was 262, and thus *n* = 8 through calculation.

The measured value of the hard segment content calculated by elemental analysis was substituted into Equation (6) [6]:(6)$$M_{\mathrm{nh}}=M_{\mathrm{ns}}\times \frac{C_{h}}{C_{s}}$$
where *M*_ns_ = 2000, *C*_h_ = 19.31, and *M*_nh_ = 478.62 after calculation. The relative molecular mass of the hard segment link was 340; thus, it could be calculated that m = 1. In the eight other polyether-based TPU structures with different design ratios, m = 2, 2, 3, 3, 3, 4, 4, and 4 as derived from this equation. The possible structural equation of the polyester-based TPUs was inferred, as shown in Figure 7.

Moreover, the synthetic chemical equation of the polyether-based TPUs using MDI and BDO as the hard segment and PTMEG as the soft segment is shown in Figure 8.

Similarly, in this study, the hydroxyl-terminated PTMEG (C_4_H_10_O_2_) with a relative molecular mass of 2000 was used. The relative molecular mass of the monomer was 90, and *n* = 22 by calculation. According to Equation (5), in the eight other polyether-based TPU structures with different design ratios, m = 2, 2, 2, 3, 3, 3, 4, 4, and 4, respectively. Combining the measured relative molecular mass values, the structural equation of the polyether-based TPUs was inferred, as shown in Figure 9.

#### 3.1.4. Nuclear Magnetic Resonance (NMR)

NMR spectroscopy is an effective method for the identification of polyurethane structure. Generally, to further confirm the sequence structure of the synthesized TPUs, the ^13^C and ^1^H NMR spectra are shown in Figure 10 and Figure 11, respectively.

Figure 10 shows the magnetic ^13^C NMR spectra of the polyester-based TPUs. The structure of the TPUs with reference to the different chemical shift (δ) values of different C atoms was analyzed. The carbonyl signal peaks appeared at both δ 148.67 and δ 171.68 ppm positions due to the formation of hydrogen bonds that likely caused the chemical shift of the carbon atoms on the carbonyl group. The peak areas can be inferred that the structures at δ 171.68 and δ 168.25 ppm were HN-C=O in the hard segment in terms of the comparison of the peak areas and δ 148.67 ppm should be O=C-O or O=C. At δ 115–140 ppm, C were on the benzene ring in MDI. However, the characteristic peaks from δ 20 to δ 40 ppm were the methylene (CH_2_) of soft segment. At δ 58–70 ppm, there might be C-O or C on the interface of the hard and soft segments. Therefore, it can be considered that the degree of phase separation of polyester-based TPUs was better [20,21].

The ^1^H NMR spectrum of polyester-based TPUs revealed the chemical resonance of aliphatic and aromatic protons, and that at δ 1.26 ppm was C-CH_2_-C while at δ 2.12 ppm was O=C-CH_2_ [22]. δ 0.87 ppm was assigned to –(CH_2_)–. Moreover, the chemical shift of CH–N appeared at δ 4.67 ppm in the hard segment. The signals at δ 7.67 and δ 7.02 ppm were the H on the benzene ring in MDI. Finally, the signal at about δ 0.5 ppm belonged to the R—OH of soft segments [38].

As shown in Figure 11, the ^13^C NMR spectra of polyether-based TPUs showed that δ 153.52 ppm was attributed to the HN-C=O in MDI. Furthermore, signals between δ 118 and 137 ppm were C on the benzene ring in MDI. In addition, δ 20 to δ 40 ppm were the methylene (CH_2_) of soft segment. The peak at δ 63.25 ppm was assigned to C-O or C in the hard and soft segments.

The ^1^H NMR spectrum of polyether-based TPUs exhibited the chemical resonance of C-CH_2_-C and O=CH_2_ at δ 3.36 and δ 9.69 ppm, and a small characteristic signal at δ 9.59 ppm can be assigned to the NH in urethane group. Besides, the chemical shift of R–NH appeared at δ 4.44 ppm in the hard segment. The signals at δ 7.67 and δ 7.02 ppm were the H on the benzene ring in MDI, whereas around δ 0.5 ppm was the R-OH of soft segment.

The ^13^C and ^1^H NMR spectra of polyester-based and polyether-based TPUs exhibited various aliphatic as well as aromatic chemical resonances corresponding to the structure of samples as labeled in the spectra [39]. It should be pointed out that the ^1^H NMR spectrum of polyester-based TPUs exhibited the chemical shifts related to PBA and MDI. In contrast, the ^1^H NMR spectrum of polyether-based TPUs exhibited the chemical shifts related to PTMEG and MDI. It can be inferred that hydrogen bonds were formed between the molecules of the TPUs, namely that both polyester-based and polyether-based TPUs contained many polar groups. As seen, all chemical shifts were labeled in the ^13^C and ^1^H NMR spectra of copolymers.

The analysis presented above proved that the desired products were obtained. Moreover, it was further proved by FTIR, EA, and GPC together that TPUs were obtained from PBA or PTMEG as soft segment, BDO as chain extender, and MDI as hard segment. Furthermore, the chemical mechanisms also proved that both polyester-based and polyether-based TPUs contained many polar groups and inferred that hydrogen bonds were formed between the molecules of the TPUs. In addition, the soft and hard segments can form microphase regions and produce microphase separation. The TPUs with a linear structure can be physically cross-linked by hydrogen bonding.

### 3.2. Thermal Characteristics

#### 3.2.1. Vicat Softening Temperature (VST)

The VST of a material is closely related to its heat tolerance, which can reflect the movement ability of chain segments. VST cannot be directly used to evaluate the actual temperature of a material, but it can be employed to guide the quality control of the material and is regarded as an evaluation standard for the heat resistance of a material [40].

Figure 12 shows that the VST of polyester-based TPUs was higher than that of the polyether-based TPUs in the same circumstance; however, the polyester-based TPUs exhibited a similar variation trend as polyether-based TPUs. When the hard segment content was identical, the VST of the TPUs gradually increased with increasing *r*. At a constant *r*, the VST of the TPUs gradually increased with the increase in the hard segment content, i.e. the hard segment content showed a more important effect on the heat resistance of TPUs as compared to *r* [41]. When the hard segment content was increased by 10%, the VST of the TPUs increased more than 45%, which meant that the TPUs exhibited a weak flexibility at high temperature and could not move freely under a constant force. Therefore, the rigidity of the TPUs increased with increasing content of the hard segment, and the VST also increased because deformation tended to be difficult.

In addition, it was found that the hard segment content remained unchanged; both the relative molecular weight and the VST tended to increase with *r* increasing. Otherwise, when the *r* was a fixed value, as the hard segment content increased, the relative molecular weight of the TPUs decreased, while the VST results still showed an upward growth trend, indicating that the relative molecular weight had a weak effect on the heat resistance of TPUs.

#### 3.2.2. Thermogravimetric (TG)

TGA and Derivation Thermogravimetry (DTA) were employed to investigate the thermal stability of polyester-based and polyether-based TPUs. As shown in Figure 13, the TG curves of all samples showed one thermal decomposition process. The weight loss of the polyester-based TPUs accounted for about 20%, 30%, and 40% of the total mass in the temperature range of 290–410 °C when the hard segment content was 20%, 30%, and 40%, respectively. The weight loss stage at 410–500 °C was attributed to the thermal cracking of the soft segment (polyester) to generate small molecular gases and macromolecular volatile components; therefore, the weight loss was most obvious at this stage. Furthermore, the residue of each sample was slowly decomposed after 500 °C. The test results show that the thermal mass loss ratio of the soft and hard segments to the total mass was consistent with the total mass ratio of the soft and hard segments of TPUs [42]. This meant that the decomposition of the weight loss temperature for TPUs occurred at about 290–410 °C owing to the hard segment thermolysis. Detailed thermal degradation parameters are shown in Table 7.

In addition, as shown in Table 7, for the identical hard segment content, *T*_5%_ and *T*_10%_ decreased with the increase of *r*. Similarly, as the hard segment content increased, when the *r* was constant, *T*_5%_ and *T*_10%_ decreased. Within a certain range of the hard segment content, *T*_5%_ and *T*_10%_ decreased with the increase of the relative molecular weight of the polyester-based TPUs. It was found through the comparison of the hard segment content and *r* that the relative molecular weight showed a slight effect on the thermal weight loss of polyester-based TPUs.

The TG curves of the polyether-based TPUs are shown in Figure 14, and the ratio of thermal decomposition temperature of polyether-based TPUs to residual mass is shown in Table 8. All polyether-based TPUs displayed distinct weight loss at two stages. The polyether-based TPUs with higher hard segment content showed lower initiation temperature and lower thermal stability [43]. At the first stage of pyrolysis, the temperatures of maximum weight loss were around 278–380 °C, which was similar to the curves of polyester-based TPUs, and this stage almost ended at 410 °C. The second stage of weight loss happened from 420 to 440 °C in all polyether-based TPUs regardless of soft segment content and molecular weight, which was caused by the thermal cracking of soft segment polyether to generate small molecular gases and macromolecular volatile components. Finally, the residue slowly decomposed after 450 °C, and then remained invariable up to 800 °C.

Table 8 shows that the thermal stability of polyester-based TPUs was obviously better than that of polyether-based TPUs. In addition, *T*_5%_, *T*_10%_
*t*_D_, and *T*_max_ of the polyether-based and polyester-based TPUs seemed to have an identical variation tendency. Nevertheless, the residual mass fraction of polyether-based TPUs was evidently smaller than that of polyester-based TPUs, which indicated that polyether-based TPUs burned more fully than polyester-based TPUs.

#### 3.2.3. Differential Scanning Calorimetric (DSC)

The crystallization of polymer is a process in which macromolecular chains transform from irregular arrangement to compact packing [44,45], based on which the crystallization status of a material can be judged by the position of the endothermic or exothermic peak in the DSC curve [46]. Generally, TPUs display a structural characteristic of separate block polymers containing soft and hard segments. The DSC test results can not only characterize the microphase separation behavior of block copolymers, but also demonstrate the glass transition temperature (*T*_g_) of TPUs. Therefore, it is of great significance to investigate the crystallization behavior of TPUs to understand the relationship between their structures and properties since the crystallization of TPUs directly affects the mixing and separation of microphases.

Figure 15 and Figure 16 show the cooling and second heating curves of polyester-based and polyether-based TPUs, respectively; their thermal transitions and relative crystallinities are shown in Table 9 and Table 10. The peak temperature of the melting peak is defined as the melting point (*T*_m_), the peak temperature of the crystallization peak is defined as the crystallization temperature (*T*_p_), and the peak areas in curves represent the crystallization enthalpy (Δ*H*_c_) and the melting enthalpy (Δ*H*_m_), respectively. In this study, both synthesized TPUs contained identical hard segment composition (MDI/BDO). To analyze the effects of the hard segment content on their crystallization, the hard segment crystallinity (*X*_hs_) of both TPUs was calculated according to the relative crystallinity equation: *X*_c_ = (Δ*H*_m_ − Δ*H*_max_)/Δ*H*_max_ × 100% (where Δ*H*_max_ is the melting enthalpy by 100%, with the melting enthalpy by 100% of hard segment being 150.6 J·g^−1^ [6]). Generally, the more orderly are the molecular chains in the synthesized TPU, the better are the symmetry and corresponding crystallinity, and the faster is the crystallization rate [47].

As shown in Figure 15 and Table 9, Δ*H*_m_ of the polyester-based TPUs increased apparently with the increase of the hard segment content. When *r* = 1, the hard segment crystallinity raised from 35.21% to 67.98%. This is because the molecular weight of the hard segment increased, and the hard segment phase exhibited high purity and preferable phase separation. With regard to the same hard segment content, the Δ*H*_m_ of polyester-based TPUs increased first and then decreased, the peak shape became wider, and the *T*_m_ and peak area became smaller with increasing *r*. Meanwhile the crystallization peak area decreased, the *T*_p_ decreased distinctly, and the peak shape widened slightly. This is mainly because the molecular weight gradually increased with the increase of *r* and the molecular weight distribution became wider.

After the introduction of many –NCO, the symmetry and regularity in the polyester-based TPU molecular chains were disrupted, the crystallization ability of TPUs was gradually weakened, and thus the crystallization ability of the material was strengthened and then weakened. In addition, the *T*_g_ gradually increased, which was mainly related to the irregular arrangement of –CH_2_ in the molecular chains. The synthesized TPUs indicated a lower *T*_g_ from −40.74 to −47.92 °C within the range of the hard segment content used. *T*_g_ exhibited a slight increase with increasing content of the hard segment; similar results were reported by Illinger et al. [48].

As can be seen clearly in Figure 16, when the hard segment content was greater than 30%, a shift peak appeared at 75–120 °C in the DSC curves; as the hard segment content increased, the shift peak gradually became apparent and toward the direction of higher temperature. Hewitt et al. [49] attributed this to the glass transition and plasticization of the hard segment phase. The occurrence of this phenomenon may also be the reason that the hard segment phase transitioned from a certain ordered state to a disordered state. As the hard segment content increased, the average length of the hard segment gradually increased, aggregation between hard segments gradually strengthened, and the ordered level increased, which suggested that, as the hard segment content in the TPUs increased the degree of both two-phase miscibility and microphase separation gradually increased. When the temperature was increased enough to destroy the aggregation force between the hard segments, the ordered state dissociated into the disordered state. It was found that the shift peak tended to be obvious and the dissociation temperature drifted to high temperature in the DSC curves. In summary, with an increase in the hard segment content, the structure of the polyester-based TPUs became more regular and easier to crystallize, and the degree of crystallinity presented basically no effect on *T*_g_, which was consistent with the findings of Seefried et al. [50].

As shown in Figure 16 and Table 10, the crystalline character of polyether-based TPUs were stronger, which had the sharp melting peak and crystallization peak, and then the rate of its crystallization was faster, which was attributed to the regular arrangement of –CH_2_ in chains of molecules inside polyether-based TPUs. When *r* = 1, the hard segment crystallinity increased from 56.28% to 59.46%, because of the increased molecular weight of the hard segment that had higher purity and better phase separation. Both Δ*H*_m_ and *X*_hc_ increased and then decreased with increasing *r*; concurrently, the backbone could also maintain a better symmetry and regularity in regard to the identical hard segment content.

In this study, the molecular weights of PBA and PTMEG were both 2000, and, even if some hard segment phase was dissolved into the soft segment phase, it had little influence on the movement ability of molecules in the whole long soft segment. In other words, the anchoring effect of the hard segment on the soft segment was not obvious. For a certain content of *r*, the melting peak strength decreased with the increase of the hard segment content, the peak shape widened, and both *T*_m_ and *T*_p_ increased, whereas the area of the melting peak and the crystalline peak tended to be smaller. *X*_hs_ gradually increased with an increase in the hard segment content, and *T*_g_ gradually decreased accordingly. PBA was used as the soft segment in the polyester-based TPUs with more hydrogen bonds, which cannot effectively form a microphase separation structure; hence, the melting shift temperature of the hard segment was lower. The incorporation of PTMEG destroyed the original orderly arrangement of molecular chains; nevertheless, it can effectively form a microphase separation structure, and thus the hard segment exhibited the highest melting transition temperature, which was consistent with the TGA findings. In addition, as the content of the hard segment increased, the *T*_g_ of polyester-based and polyether-based TPUs showed an upward variation trend as result of the higher hard segment content, the lower molecular weight, and the greater rigidity of the molecular chains. The relative crystallinity of the hard segment of the polyester-based TPUs in the same conditions was significantly higher than that of the polyether-based TPUs, and the *T*_g_ of the polyether-based TPUs was evidently lower than that of the polyester-based TPUs. This meant that the thermal behavior of polyester-based TPUs was better than that of the polyether-based TPUs. However, the low temperature performance of polyether-based TPUs was superior.

Thermal characteristics analysis results show that the hard segment content and *r* of the polyester-based and polyether-based TPUs had a significant effect on the heat resistance. When the hard segment content was increased by 10%, the VST of the TPUs grew by more than 45%. However, the relative molecular weight exhibited a weak effect on the heat resistance of TPUs. Moreover, the thermal stability of polyester-based TPUs was obviously better than that of polyether-based TPUs, hence the polyether-based TPUs burned fully more than the polyester-based TPUs. The higher was the hard segment content, the lower was the molecular weight, and the greater was the rigidity of the molecular chains. The relative crystallinity of the hard segment of the polyester-based TPUs under the same conditions was significantly higher than that of the polyether-based TPUs, and the polyether-based TPUs had a distinctly lower *T*_g_ than the polyester-based TPUs.

### 3.3. Physical and Mechanical Properties

Table 11 shows that, when the hard segment content was kept constant, the relative molecular weight of polyester-based TPUs increased as *r* increased. The shore hardness, tensile strength, tear strength, and 300% elongation stress showed an upward trend. However, the test results of the impact strength and elongation at break show a downward trend. When *r* was constant, the relative molecular weight of polyester-based TPUs decreased with the increase of the hard segment content. The shore hardness, tensile strength, tear strength, and 300% elongation stress all showed an upward trend, while the impact strength and elongation at break showed a downward trend. It was demonstrated that the relative molecular weight is not an important factor affecting the mechanical properties of polyester-based TPUs.

As shown in Table 12, when the hard segment content was 20%, no models of pendulums could fracture the test samples due to the larger toughness of the material. The performance index of polyether-based TPUs exhibited a good consistency with that of polyester-based TPUs. This was mainly attributed to the increased hydrogen bonding between hard segments and intermolecular forces with an increase in the hard segment content and cohesive energy density, leading to the increased tensile strength, whereas the elongation at break decreased due to the decreased soft segment content in the material, and, correspondingly, the freely rotatable segments in the molecular chain were reduced. The increase of the shore hardness might be due to the increased polar group content such as urethane in the polyurethane molecule, the crosslinking density, and the rigidity. The difference of the mechanical properties between polyester-based and polyether-based TPUs in shown in Table 13.

With identical hard segment content, as r increased, the polyester-based TPUs showed similar physical and mechanical properties as the polyether-based TPUs, namely the tensile strength increased significantly, while the elongation at break decreased. This is because, with *r* > 1, the excessive isocyanate group reacted with the hydrogen atom on the carbamate group, resulting in the slight cross-linking of the polyurethane molecular chain, increased tensile strength, and reduced elongation at break. However, polyester-based TPUs performed better than polyether-based TPUs as result of the polyester polyols containing more polar ester groups [51,52,53]. The shore hardness, tensile strength, and tear strength were increased due to the presence of ester groups increasing the intermolecular forces. Although the number of ester groups increased, the ordered structure of the backbone in TPUs could be reduced, hence the impact strength and fracture elongation decreased.

In addition, it was found that the relative molecular mass of TPUs presented a weak effect on the mechanical properties, whereas the crystallinity of the hard segment exhibited a significant effect on the properties of TPUs. The crystallinity of polyester-based TPUs was 4.64–67.98%, shore hardness was 48–95 A, tensile strength was 8.28–36.2 MPa, tear strength was 67.2–115.6 kN·m^−1^, and the tensile stress at 300% was 2.71–9.15 MPa. By contrast, the crystallinity of polyether-based TPUs was 48.41–59.46%, shore hardness was 39–85 A, tensile strength was 5.48–22.76 MPa, tear strength was 20.2–52.1 kN·m^−1^, and the tensile stress at 300% was 0.28–1.51 MPa. The reason for this phenomenon is that the hard segment of polyester-based TPUs formed multiple small hard segments with a larger surface area than the polyether-based TPUs, which could effectively restrict the deformation of the soft segment substrate and prevent the growth of cracks. Therefore, the hardness, tear strength, and tensile strength of polyester-based TPUs were higher than those of polyether-based TPUs [6].

Thus, based on the physical and mechanical properties, the intermolecular force increased owing to the large number of polar groups (ester groups) in the polyester polyol molecular chain and the formed hydrogen bonds between the soft segment and the hard segment. Every part of the soft segment phase formed a hydrogen bond with the hard segment, which increased the compatibility between the soft segment and the hard segment in the polyester-based TPUs. The hard segment phase was more evenly distributed in the soft segment phase of the polyester-based TPUs, which made the degree of microphase separation lower than that of the polyether-based TPUs. Therefore, the polyester-based TPUs had greater internal heat generation, dynamic characteristics, and low temperature resistance worse than the polyether-based TPUs. This was because polyester polyols contained many highly polar ester groups, which not only resulted in hydrogen bonds formation between the hard segments, but also the polar groups on the soft segments could form hydrogen bonds with the hard segments, leading to the increased intermolecular force, and the hard segment phase was more uniformly distributed in the soft segment phase, which increased the physical crosslinking points, and then some polyester polyols can form soft segment crystals. Consequently, the polyester-based TPUs exhibited higher hardness, strength, and heat-resistant oxygen aging performance than the polyether-based TPUs.

## 4. Conclusions

In this study, polyester-based and polyether-based TPU modifiers were successfully synthesized by the semi-prepolymer method. Then, the chemical mechanisms, thermal characteristics, and mechanical properties were investigated. The following conclusions were drawn from the results of this study. (1)The polyester-based and polyether-based TPUs with the hard segment content (r for each hard segment content was designed as 0.95, 1, and 1.05, respectively) of 20%, 30%, and 40% were successfully synthesized by the semi-prepolymer method. The linear structure of TPUs with different hard segment contents were depicted.(2)The hard segment contents and *r* of the polyester-based and polyether-based TPUs exhibited a significant effect on the heat resistance. For the identical hard segment contents and r, the polyester-based TPUs had obviously higher thermal stability than the polyether-based TPUs. However, the relative crystallinity of the hard segment of the polyester-based TPUs under the same conditions was significantly higher than that of the polyether-based TPUs, and the polyether-based TPUs had a distinctly lower *T*g than the polyester-based TPUs.(3)The relative molecular mass showed a weak effect on the mechanical properties, whereas the crystallinity of hard segment presented a pronounced effect on the properties in terms of the synthetized TPUs. The crystallinity of the polyester-based TPUs was 4.64–67.98%, shore hardness was 48–95 A, tensile strength was 8.28–36.2 MPa, tear strength was 67.2–115.6 kN·m^−1^, and the tensile stress at 300% was 2.71–9.15 MPa. The crystallinity of the polyether-based TPUs was 48.41–59.46%, shore hardness was 39–85 A, tensile strength was 5.48–22.76 MPa, tear strength was 20.2–52.1 kN·m^−1^, and the tensile stress at 300% was 0.28–1.51 MPa.(4)The average structural parameters of the TPUs with different hard segments and r could be obtained by combining FTIR, EA, and GPC. The mechanisms and the chemical structure of TPUs could be inferred by 1H and 13C NMR. For the identical hard segment and r, the polyester-based TPUs had better high temperature performance, while the low temperature performance of polyether-based TPUs was better. In addition, the relative molecular mass of TPUs manifested a weak effect on both the thermal characteristics and the mechanical properties. As indicated above, the chemical structure of the designed modifiers can be accurately speculated in this study, which can provide a new idea for the research and development of asphalt modifiers.

## Figures and Tables

**Figure 1 materials-13-04991-f001:**
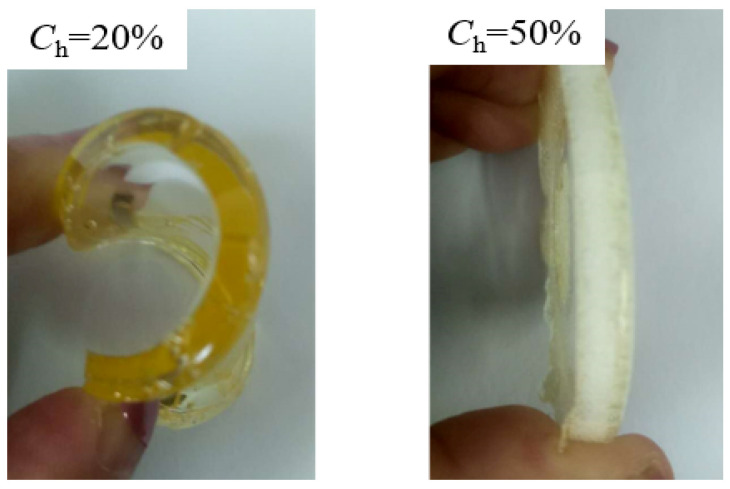
The bending of TPUs with different hard segment contents.

**Figure 2 materials-13-04991-f002:**
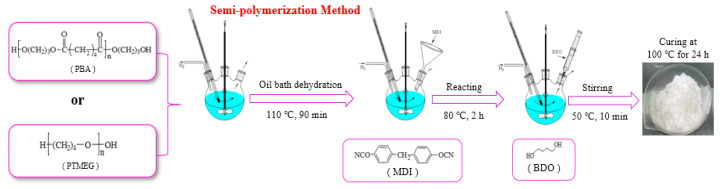
The preparation process of TPUs.

**Figure 3 materials-13-04991-f003:**
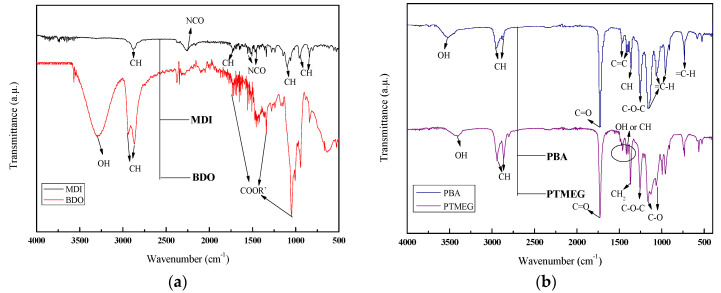
FTIR spectra of raw materials. (**a**) hard segment; (**b**) soft segment.

**Figure 4 materials-13-04991-f004:**
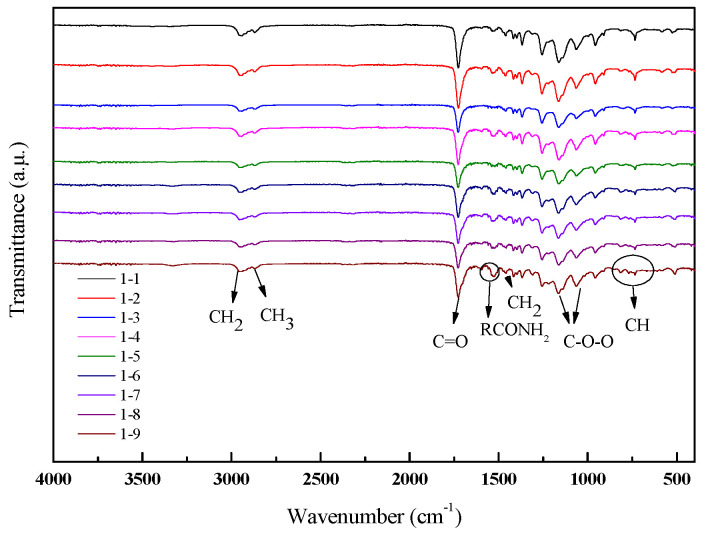
FTIR spectra of polyester-based TPUs.

**Figure 5 materials-13-04991-f005:**
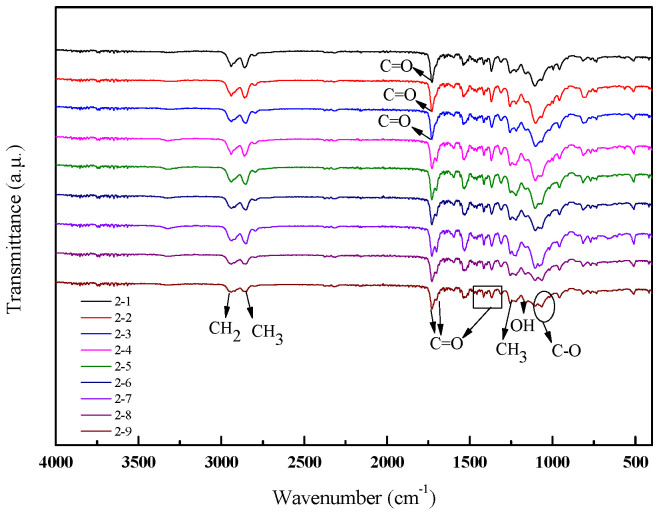
FTIR spectra of polyether-based TPUs.

**Figure 6 materials-13-04991-f006:**
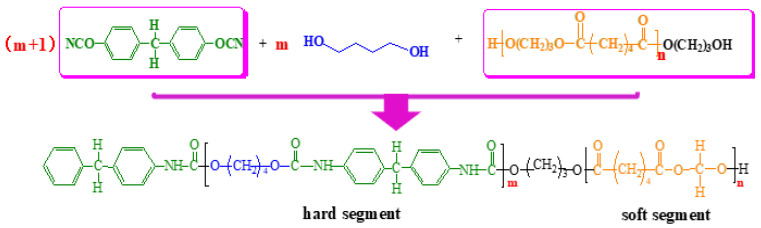
Chemical equation of MDI, BDO, and PBA synthetize polyester-based TPUs.

**Figure 7 materials-13-04991-f007:**
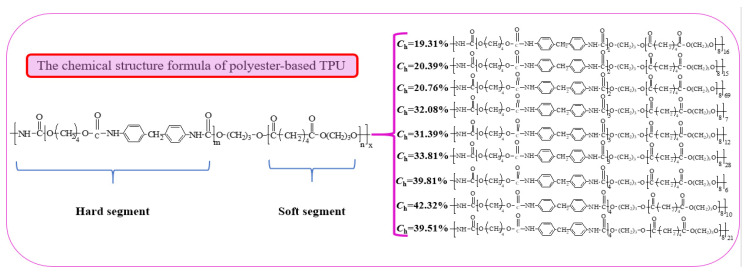
Chemical equation of polyester-based TPUs.

**Figure 8 materials-13-04991-f008:**
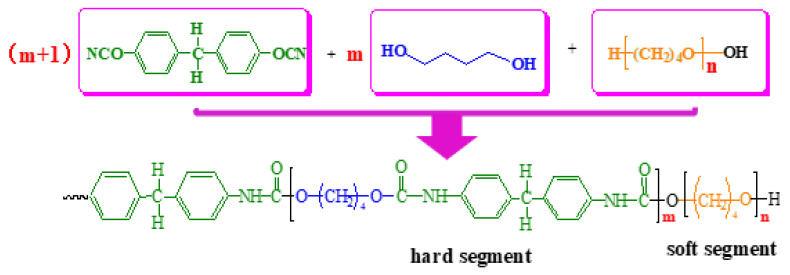
Chemical equation of MDI, BDO and PTMEG synthetize polyether-based TPUs.

**Figure 9 materials-13-04991-f009:**
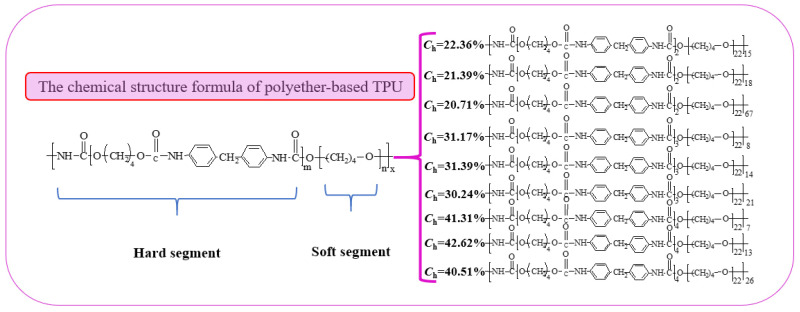
Chemical equation of polyether-based TPUs.

**Figure 10 materials-13-04991-f010:**
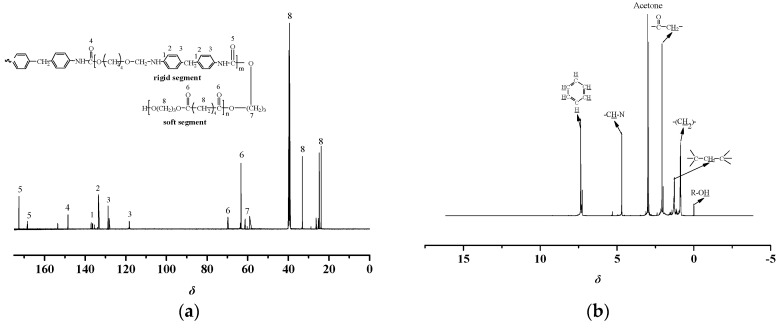
^13^C and ^1^H NMR spectra of polyester-based TPUs. (**a**) ^13^C NMR spectra; (**b**) ^1^H NMR spectra.

**Figure 11 materials-13-04991-f011:**
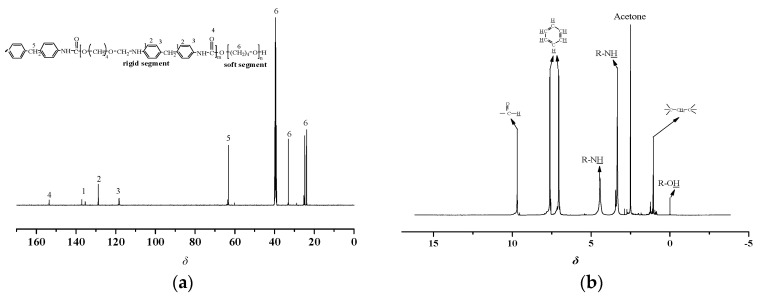
^1^^3^C and ^1^H NMR spectra of polyether-based TPUs. (**a**) ^13^C NMR spectra; (**b**) ^1^H NMR spectra.

**Figure 12 materials-13-04991-f012:**
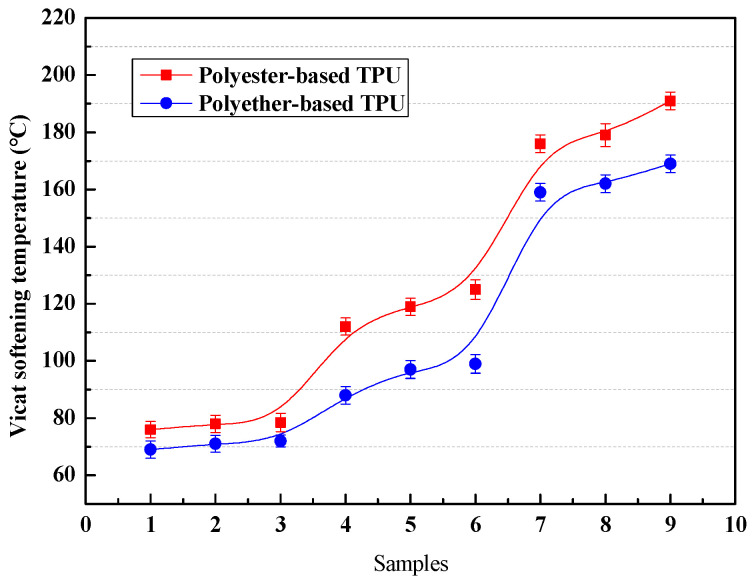
VST of polyester-based and polyether-based TPUs.

**Figure 13 materials-13-04991-f013:**
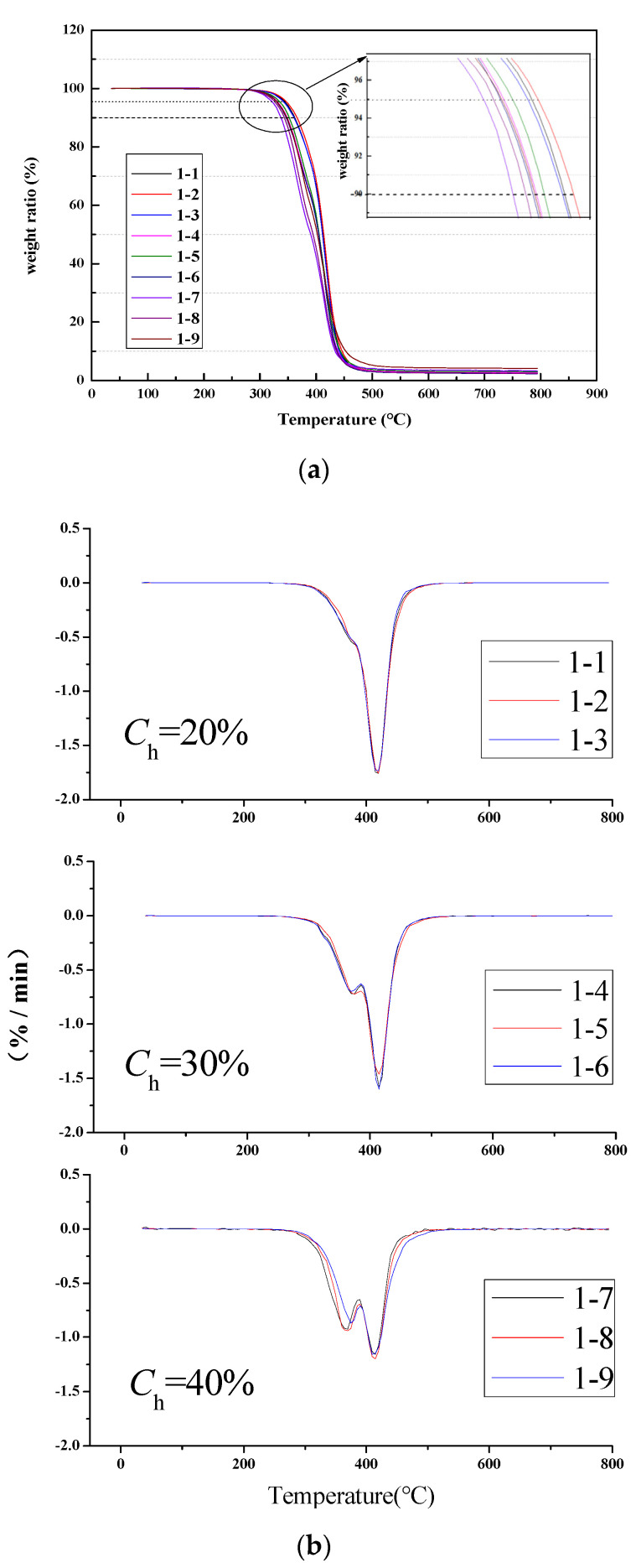
TGA and DTA curves of polyester-based TPUs. (**a**) TGA; (**b**) DTA.

**Figure 14 materials-13-04991-f014:**
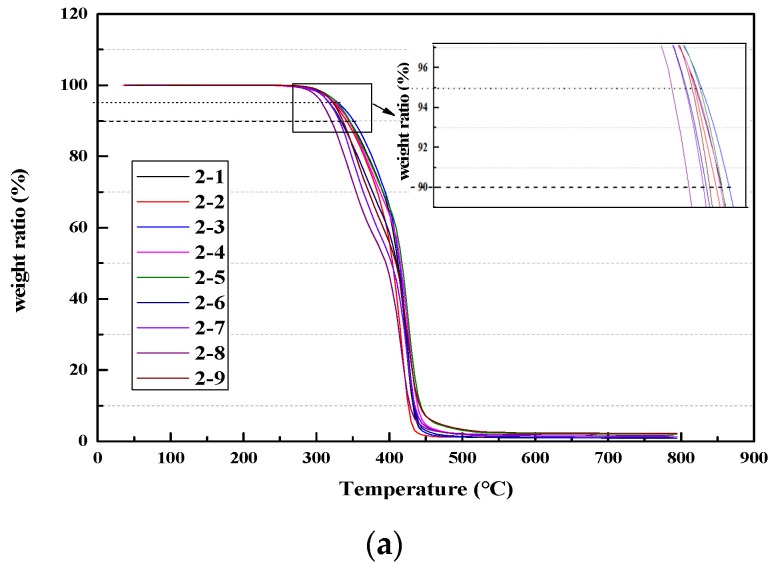
TGA and DTA curves of polyether-based TPUs. (**a**) TGA; (**b**) DTA.

**Figure 15 materials-13-04991-f015:**
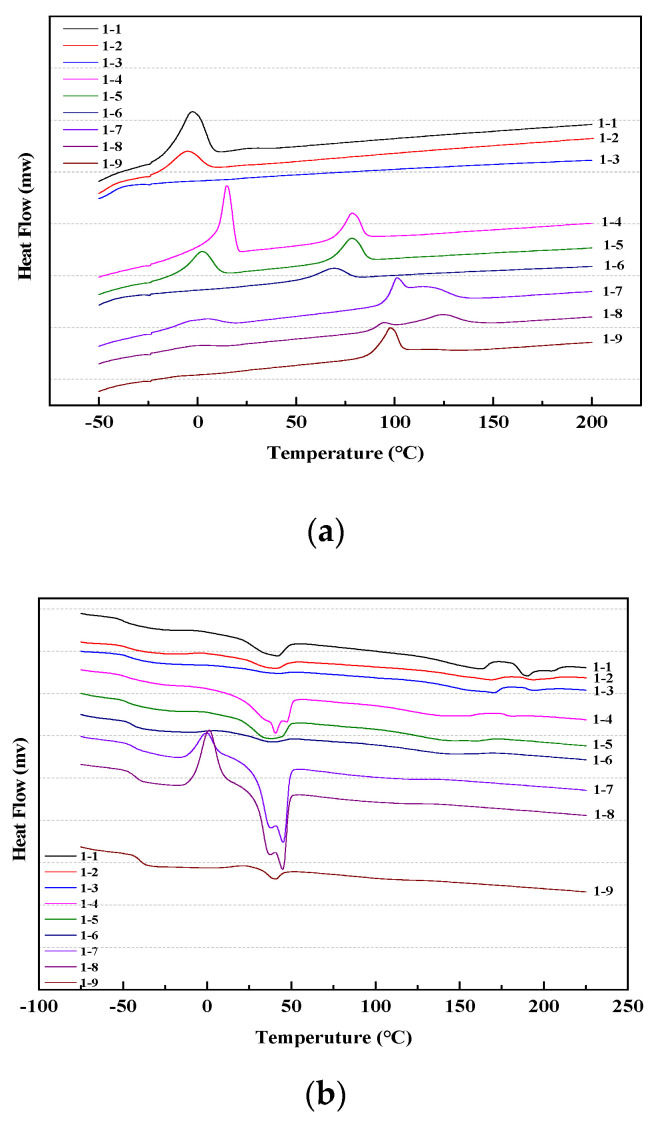
DSC curves shape of polyester-based TPUs. (**a**) Cooling curves; (**b**) Second heating curves.

**Figure 16 materials-13-04991-f016:**
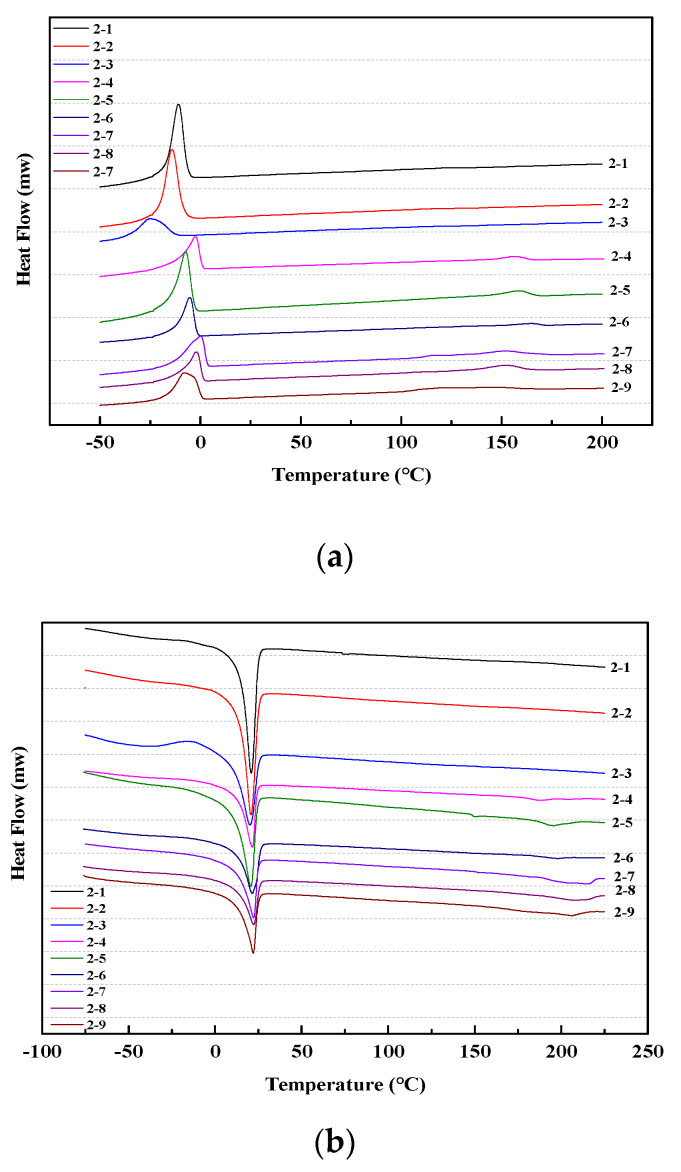
DSC curves shape of polyether-based TPUs. (**a**) Cooling curves; (**b**) Second heating curves.

**Table 1 materials-13-04991-t001:** Composition and hard segment content of polyester-based TPUs.

No.	Hard Segment Content	r	The Mass of PBA (g)	The Mass of MDI (g)	The Mass of BDO (g)
1-1	*C*_h_ = 20%	0.95	100	21.39	3.607
1-2	*C*_h_ = 20%	1	100	21.69	3.309
1-3	*C*_h_ = 20%	1.05	100	21.97	3.032
1-4	*C*_h_ = 30%	0.95	100	34.343	8.517
1-5	*C*_h_ = 30%	1	100	34.821	8.036
1-6	*C*_h_ = 30%	1.05	100	35.226	7.591
1-7	*C*_h_ = 40%	0.95	100	51.606	15.056
1-8	*C*_h_ = 40%	1	100	52.325	14.337
1-9	*C*_h_ = 40%	1.05	100	52.991	13.669

**Table 2 materials-13-04991-t002:** Composition and hard segment content of polyether-based TPUs.

No.	Hard Segment Content	r	The Mass of PTMEG (g)	The Mass of MDI (g)	The Mass of BDO (g)
2-1	*C*_h_ = 20%	0.95	100	21.39	3.607
2-2	*C*_h_ = 20%	1	100	21.69	3.309
2-3	*C*_h_ = 20%	1.05	100	21.97	3.032
2-4	*C*_h_ = 30%	0.95	100	34.343	8.517
2-5	*C*_h_ = 30%	1	100	34.821	8.036
2-6	*C*_h_ = 30%	1.05	100	35.226	7.591
2-7	*C*_h_ = 40%	0.95	100	51.606	15.056
2-8	*C*_h_ = 40%	1	100	52.325	14.337
2-9	*C*_h_ = 40%	1.05	100	52.991	13.669

**Table 3 materials-13-04991-t003:** Element analysis of synthetic polyester-based TPUs.

Design Value of Ch (%)	*r*	C (%)	H (%)	N (%)	O (%)	Measured Value of Ch (%)
20%	0.95	66.98	3.304	1.597	28.119	19.31
20%	1	66.45	3.364	1.682	28.504	20.39
20%	1.05	66.88	3.593	1.711	27.816	20.76
30%	0.95	64.32	5.021	2.643	28.016	32.08
30%	1	63.25	5.349	2.587	28.814	31.39
30%	1.05	62.11	6.271	2.786	28.833	33.81
40%	0.95	60.23	6.234	3.281	30.255	39.81
40%	1	61.22	6.505	3.487	28.788	42.32
40%	1.05	60.04	6.387	3.256	30.317	39.51

**Table 4 materials-13-04991-t004:** Element analysis of polyether-based TPUs.

Design Value of Ch (%)	*r*	C (%)	H (%)	N (%)	O (%)	Measured Value of Ch (%)
20%	0.95	67.026	3.499	1.842	27.633	22.36
20%	1	66.496	3.526	1.763	28.215	21.39
20%	1.05	66.926	3.585	1.707	27.782	20.71
30%	0.95	64.364	4.879	2.568	28.189	31.17
30%	1	63.294	5.140	2.587	28.979	31.39
30%	1.05	62.153	5.231	2.491	30.125	30.24
40%	0.95	60.271	6.468	3.404	29.857	41.31
40%	1	61.262	7.024	3.512	28.202	42.62
40%	1.05	60.081	7.001	3.338	29.580	40.51

**Table 5 materials-13-04991-t005:** Influence of different hard segment content and r on molecular weight of polyester-based TPUs.

Samples	Hard Segment Content	*r*	${\overset{¯¯}{M}}_{n}$	${\overset{¯¯}{M}}_{w}$	${\overset{¯¯}{M}}_{z}$	${\overset{¯¯}{M}}_{v}$	D= $\left({\overset{¯¯}{M}}_{w}/{\overset{¯¯}{M}}_{n}\right)$
1-1	*C*_h_ = 20%	0.95	31,464	60,220	107,687	54,830	1.91
1-2	*C*_h_ = 20%	1	34,320	62,763	106,643	57,584	1.83
1-3	*C*_h_ = 20%	1.05	161,147	486,596	1,238,382	414,624	3.02
1-4	*C*_h_ = 30%	0.95	19,921	35,867	59,226	33,008	1.80
1-5	*C*_h_ = 30%	1	33,503	65,132	114,547	59,330	1.94
1-6	*C*_h_ = 30%	1.05	78,214	222,388	566,171	190,372	2.84
1-7	*C*_h_ = 40%	0.95	18,038	32,361	53,800	29,759	1.79
1-8	*C*_h_ = 40%	1	29,844	58,710	104,149	53,367	1.97
1-9	*C*_h_ = 40%	1.05	64,610	146,932	290,382	131,132	2.27

**Table 6 materials-13-04991-t006:** Influence of different hard segment content and r on molecular weight of polyether-based TPUs.

Samples	Hard Segment Content	*r*	${\overset{¯¯}{M}}_{n}$	${\overset{¯¯}{M}}_{w}$	${\overset{¯¯}{M}}_{z}$	${\overset{¯¯}{M}}_{v}$	D= $\left({\overset{¯¯}{M}}_{w}/{\overset{¯¯}{M}}_{n}\right)$
2-1	*C*_h_ = 20%	0.95	35,049	66,330	115,125	60,622	1.89
2-2	*C*_h_ = 20%	1	42,159	82,033	145,939	74,669	1.95
2-3	*C*_h_ = 20%	1.05	157,362	510,111	1,484,062	426,633	3.24
2-4	*C*_h_ = 30%	0.95	22,457	42,907	73,531	39,225	1.91
2-5	*C*_h_ = 30%	1	38,713	88,280	178,185	78,512	2.28
2-6	*C*_h_ = 30%	1.05	58,642	138,433	292,275	122,292	2.36
2-7	*C*_h_ = 40%	0.95	23,202	47,824	85,831	43,334	2.06
2-8	*C*_h_ = 40%	1	39,957	93,278	187,658	822,903	2.33
2-9	*C*_h_ = 40%	1.05	81,367	286,026	924,066	234,759	3.51

**Table 7 materials-13-04991-t007:** Degradation temperature and char residue of polyester-based TPUs.

Samples	*T*_5%_^a^ (°C)	*T*_10%_^b^ (°C)	*t*_D_^c^ (°C)	*T*_max_^d^ (°C)	Char Residue (%)
1-1	350.7	359.5	354.7	414.3	2.43
1-2	349.5	364.6	358.6	419.4	3.07
1-3	344.6	354.3	346.4	414.3	2.39
1-4	333.6	347.6	341.3	414.5	2.24
1-5	334.8	354.1	352.8	419.5	3.01
1-6	329.5	345.9	340.9	414.2	2.43
1-7	324.6	344.5	331.2	413.9	2.44
1-8	329.2	347.1	334.7	414.6	3.39
1-9	328.1	336.2	335.9	413.7	4.09

^a^ Temperature corresponding to 5% weight loss; ^b^ temperature corresponding to 10% weight loss; ^c^ extrapolating the onset temperature; ^d^ temperature of maximum weight loss percent.

**Table 8 materials-13-04991-t008:** Degradation temperature and char residue of polyether-based TPUs.

Samples	*T*_5%_^a^ (°C)	*T*_10%_^b^ (°C)	*t*_D_^c^ (°C)	*T*_max_^d^ (°C)	Char Residue (%)
2-1	324.5	334.9	329.6	429.5	0.79
2-2	324.7	340.7	334.2	419.5	0.92
2-3	329.7	350.6	339.9	424.6	0.89
2-4	325.5	334.1	330.2	419.4	1.28
2-5	327.1	345.8	331.7	424.8	0.91
2-6	317.9	333.6	329.8	420.5	1.32
2-7	308.8	329.6	289.4	419.4	1.33
2-8	306.3	319.7	284.7	419.2	1.65
2-9	320.1	335.5	289.6	424.9	2.21

^a^ Temperature corresponding to 5% weight loss; ^b^ temperature corresponding to 10% weight loss; ^c^ extrapolating the onset temperature; ^d^ temperature of maximum weight loss percent.

**Table 9 materials-13-04991-t009:** Thermal transitions and relative crystallinity of polyester-based TPUs.

Samples	Δ*H*_m_ (J·g^−^^1^)	*T*_m_ (°C)	Δ*H*_c_ (J·g^−1^)	*T*_p_ (°C)	*X*_hs_ (%)	*T*_g_ (°C)
1-1	44.67	48.1	7.18	2.3	24.89	−47.9
1-2	63.67	48.4	10.64	4.6	35.21	−47.2
1-3	14.18	43.8	7.19	7.9	4.64	−43.6
1-4	155.06	45.1	68.37	−2.5	57.56	−44.5
1-5	157.01	45.9	55.97	−5.2	67.09	−43.9
1-6	39.35	43.8	13.57	−6.3	17.11	−41.8
1-7	119.47	46.6	31.42	14.7	58.47	−42.4
1-8	123.67	47.9	21.28	2.4	67.98	−42.1
1-9	37.91	42.9	9.96	1.5	18.56	−40.7

**Table 10 materials-13-04991-t010:** Thermal transitions and relative crystallinity of polyether-based TPUs.

Samples	Δ*H*_m_ (J·g^−1^)	*T*_m_ (°C)	Δ*H*_c_ (J·g^−1^)	*T*_p_ (°C)	*X*_hs_ (%)	*T*_g_ (°C)
2-1	162.62	24.86	89.69	−10.99	48.41	−54.7
2-2	171.92	25.68	87.16	−14.31	56.28	−52.6
2-3	148.67	21.98	68.95	−24.98	52.93	−48.3
2-4	143.95	25.49	69.18	−5.46	49.65	−52.9
2-5	157.57	26.54	70.99	−7.40	57.49	−50.4
2-6	139.34	22.26	58.99	−17.25	53.35	−47.7
2-7	139.29	25.76	57.61	0.02	54.24	−51.9
2-8	146.65	26.93	57.10	−2.23	59.46	−49.6
2-9	143.11	23.32	62.29	−8.02	53.67	−46.2

**Table 11 materials-13-04991-t011:** Physical and mechanical properties of polyester-based TPUs.

Samples	Hard Segment	*r*	Shore Hardness (A)	Impact Strength (kJ/m^2^)	Tensile Strength (MPa)	Tear Strength (kN·m^−1^)	Stress at 300% Strain (MPa)	Break Elongation (%)
1-1	*C*_h_ = 20%	0.95	48	17.6	8.25	67.2	2.71	740
1-2	*C*_h_ = 20%	1	51	17.3	8.37	68.1	3.05	732
1-3	*C*_h_ = 20%	1.05	52	16.9	8.98	68.9	3.39	728
1-4	*C*_h_ = 30%	0.95	81	15.3	25.82	90.1	5.26	581
1-5	*C*_h_ = 30%	1	82	15.3	26.51	90.4	5.85	575
1-6	*C*_h_ = 30%	1.05	85	14.7	27.09	91.6	5.97	579
1-7	*C*_h_ = 40%	0.95	91	9.6	35.41	109.1	8.04	441
1-8	*C*_h_ = 40%	1	93	8.7	35.72	114.3	8.73	435
1-9	*C*_h_ = 40%	1.05	95	8.5	36.32	115.6	9.15	407

**Table 12 materials-13-04991-t012:** Physical and mechanical properties of polyether-based TPUs.

Samples	Hard Segment	*r*	Shore Hardness (A)	Impact Strength (kJ/m^2^)	Tensile Strength (MPa)	Tear Strength (kN·m^−1^)	Stress at 300% Strain (MPa)	Break Elongation (%)
2-1	*C*_h_ = 20%	0.95	39	-	5.48	20.2	0.28	1258
2-2	*C*_h_ = 20%	1	40	-	6.21	20.9	0.39	1137
2-3	*C*_h_ = 20%	1.05	40	-	6.44	21.6	0.40	1000
2-4	*C*_h_ = 30%	0.95	64	27.4	8.53	32.7	0.77	900
2-5	*C*_h_ = 30%	1	66	23.5	9.67	34.9	0.86	916
2-6	*C*_h_ = 30%	1.05	69	21.9	9.10	38	0.95	854
2-7	*C*_h_ = 40%	0.95	79	18.9	21.34	49.8	1.27	769
2-8	*C*_h_ = 40%	1	83	17.3	22.15	49.8	1.39	747
2-9	*C*_h_ = 40%	1.05	85	16.2	22.76	52.1	1.51	739

**Table 13 materials-13-04991-t013:** Performance comparison of polyester-based and polyether-based TPUs.

Testing Items	Performance Comparison
Shore hardness	polyester-based TPUs > polyether-based TPUs
Impact strength	polyester-based TPUs < polyether-based TPUs
Tensile strength	polyester-based TPUs > polyether-based TPUs
Tear strength	polyester-based TPUs > polyether-based TPUs
Tensile stress at 300%	polyester-based TPUs > polyether-based TPUs
Elongation at break	polyester-based TPUs < polyether-based TPUs

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
