# Peer review of "Design and Performance of Polyurethane Elastomers Composed with Different Soft Segments"

_materials, 2020, doi:10.3390/ma13214991_

Round 1

Reviewer 1 Report

It seems to be that work entitled “Study on synthesis and properties of thermoplastic polyurethane elastomers as asphalt modifier based on microstructure” will present the results focused on the effect of thermoplastic polyurethane elastomers (TPU) addition on the asphalt properties. There is any information about the effect of TPU materials on asphalt properties in the introduction as well as in the experimental part. This paper is rather about TPU materials obtained using standard petrochemical monomers.

In my opinion, this work needs a major revision.

Some suggestions / comments:

  1. Change the title of this work or prove that obtained materials can be or were applied as a modifier of asphalt.
  2. Add some literature related to the application of TPE for asphalt modification and some results.
  3. Justify the selection of the monomers for TPU preparation destined for asphalt modification.
  4. Line 122: There is “The TPU can be plasticized by heating and the solvent can be dissolved” Truly I do not understand what the Authors meant?
  5. FTIR spectra of raw materials should be a part of supplementary materials.
  6. How did you verify that obtained materials can be used as an asphalt modifier?

In the text are some spelling mistakes e.g. Line 674 shore hardness – should be Shore hardness

Reviewer 2 Report

This paper presents an interesting study on the properties of thermoplastic polyurethane-based elastomers used as asphalt modifiers. This paper is scientifically sounds, and conclusions are supported by valid experimentation and results. 

A minor revision is needed in the "Abstract" section, specifically lines 17-20 that need to be rewritten.  

More detailed please see attachment.

Round 2

Reviewer 1 Report

After revision, the paper gained a new quality, but still, I cannot find the justification for the usage of TPU as an asphalt modifier. The information about TPU as an asphalt modifier you can include in the conclusion (as one of the possible ways for TPU application),  but not in the abstract. In my opinion, the abstract should be rewritten. What is more, some spelling mistakes still are in the text.
